# Associations of Traditionally Determined Left Ventricular Mass Indices and Hemodynamic and Non-Hemodynamic Components of Cardiac Remodeling with Diastolic and Systolic Function in Patients with Chronic Kidney Disease

**DOI:** 10.3390/jcm12134211

**Published:** 2023-06-22

**Authors:** Hon-Chun Hsu, Grace Tade, Chanel Robinson, Noluntu Dlongolo, Gloria Teckie, Ahmed Solomon, Angela Jill Woodiwiss, Patrick Hector Dessein

**Affiliations:** 1Cardiovascular Pathophysiology and Genomics Research Unit, School of Physiology, Faculty of Health Sciences, University of the Witwatersrand, Johannesburg 2193, South Africa; kengjulin@yahoo.com.au (H.-C.H.); grace.tade@hotmail.com (G.T.); chanelgr@gmail.com (C.R.); angela.woodiwiss@wits.ac.za (A.J.W.); 2Nephrology Unit, Milpark Hospital, Johannesburg 2193, South Africa; 3Rheumatology Unit, Rosebank Hospital, Johannesburg 2193, South Africa; 4Division of Nephrology, Department of Medicine, Chris Hani Baragwanath Hospital and Faculty of Health Sciences, University of Witwatersrand, Johannesburg 2193, South Africa; gteckie@hotmail.com; 5Internal Medicine Department, University of the Witwatersrand, Johannesburg 2193, South Africa; ahmed.solomon@icloud.com

**Keywords:** chronic kidney disease, traditionally determined left ventricular mass index, hemodynamic, non-hemodynamic cardiac remodeling components, diastolic–systolic function

## Abstract

We aimed to evaluate the extent to which different left ventricular mass parameters are associated with left ventricular function in chronic kidney disease (CKD) patients. We compared the associations between traditionally determined left ventricular mass indices (LVMIs) and hemodynamic (predicted LVMIs) and non-hemodynamic remodeling parameters with left ventricular function in patients with CKD; non-hemodynamic remodeling was represented by inappropriate left ventricular mass and inappropriate excess LVMIs (traditionally determined LVMIs-predicted LVMIs). Non-hemodynamic left ventricular remodeling parameters were strongly associated with impaired left ventricular systolic function (*p* < 0.001), whereas hemodynamic left ventricular remodeling was also related strongly (*p* < 0.001) but directly to left ventricular systolic function. Independent of one another, hemodynamic and non-hemodynamic left ventricular remodeling had associations in opposite directions to left ventricular systolic function and was associated directly with traditionally determined left ventricular mas indices (*p* < 0.001 for all relationships). Non-hemodynamic cardiac remodeling parameters discriminated more effectively than traditionally determined LVMIs between patients with and without reduced ejection fraction (*p* < 0.04 for comparison). Left ventricular mass parameters were unrelated to impaired diastolic function in patients with CKD. Traditionally determined LVMIs are less strongly associated with impaired systolic function than non-hemodynamic remodeling parameters (*p* < 0.04–0.01 for comparisons) because they represent both adaptive or compensatory and non-hemodynamic cardiac remodeling.

## 1. Introduction

Uremic cardiomyopathy affects up to 74% of patients with chronic kidney disease [1]. Uremic cardiomyopathy comprises diastolic dysfunction, underlying fibrosis and left ventricular hypertrophy [2,3]. The pathophysiological mechanisms that link chronic kidney disease with uremic cardiomyopathy include increased pre-load mediated by volume overload and anaemia and increased afterload due to arteriosclerosis and hypertension, as well as a wide range of biochemical factors, including chronic kidney disease bone and mineral disorder, renin–angiotensin–aldosterone and sympathetic nervous system overactivity, transforming growth-β, uremic toxins and endogenous cardiotonic steroids [1,2,3,4]. Diastolic dysfunction that can lead to heart failure with preserved ejection fraction is the most prevalent cardiac function abnormality in patients with chronic kidney disease [5]. In addition, ~30% of patients with end-stage chronic kidney disease experience impaired cardiac systolic function that is most strongly associated with incident mortality [5,6]. Uremic cardiomyopathy accounts for most of the markedly increased cardiovascular disease event rates in patients with chronic kidney disease [1,2,3].

Contemporary knowledge suggests that left ventricular hypertrophy typically develops in response to increased hemodynamic load, including that caused by enhanced pressure (afterload) and/or volume (pre-load) [7]. In this regard, the law of LaPlace states that the load on any myocardial region is given as pressure × radius/2 × wall thickness. Therefore, increased pressure load will cause compensatory and adaptive left ventricular concentric hypertrophy with thickening of the ventricular wall. This process occurs to prevent an increase in wall stress and thereby maintain normal ejection fraction [7]. On the other hand, volume overload causes compensatory and adaptive left ventricular eccentric hypertrophy with ventricular cavity dilatation and a decrease in the wall thickness–chamber dimension ratio. This event occurs in order to meet the demand to sustain a large stroke volume [6]. Nevertheless, we recently reported that in persons with volume-dependent hypertension, both pressure and volume load are associated to a similar extent with concentric and eccentric left ventricular hypertrophy [8].

Traditionally determined left ventricular mass indices strongly predicts cardiovascular events independent of conventional cardiovascular risk factors and coronary artery disease [9,10,11,12]. This fact has been amply reported in population studies [9] and patients with hypertension [10]. The development of left ventricular hypertrophy in response to increased hemodynamic load was well documented by de Simone et al. [13]. However, 14 to 29% of individuals experience left ventricular hypertrophy that extends beyond hemodynamic needs [14,15]. Interestingly, this fact applies to hypertensive subjects both with and without traditionally determined left ventricular hypertrophy [16]. Accordingly, in 1999, a group of investigators from Weill Medical College of Cornell University in New York, USA, introduced the entity of inappropriate left ventricular mass [17]. It was defined as the ratio of traditionally determined or measured left ventricular mass to predicted left ventricular mass based on age, sex, body size and cardiac workload. Inappropriate left ventricular mass, therefore, represents a marker of non-hemodynamic left ventricular remodeling. Subsequent studies reported that inappropriate left ventricular mass is more strongly associated with cardiovascular disease than traditionally determined left ventricular mass indices [18,19].

Very recently, and to gain further insights into the pathophysiology of heart disease among patients with primary aldosteronism, several investigators employed the novel and promising parameter-inappropriate excess left ventricular mass index [20,21,22]. This parameter is calculated as the difference between traditionally determined left ventricular mass index and predicted left ventricular mass index.

Inappropriate left ventricular mass occurs in as much as 40 to 70% of patients with chronic kidney disease [23,24], in whom it is independently associated with incident cardiovascular events [24]. Both traditionally determined left ventricular mass indices and inappropriate left ventricular mass are associated with impaired diastolic and systolic function in patients with hypertension [25]. However, whether this fact is also the case in patients with chronic kidney disease is uncertain. This hypothesis was assessed in the present study. Impaired left ventricular function is predictive of incidents of heart failure [26].

Conceptually, hemodynamic compensatory and adaptive left ventricular remodeling, as represented by predicted left ventricular mass index, as well as non-hemodynamic left ventricular remodeling as identified by inappropriate excess left ventricular mass indices, would be expected to contribute to traditionally determined left ventricular mass indices. We, therefore, hypothesized that compared to inappropriate left ventricular mass and inappropriate excess left ventricular mass indices, traditionally determined left ventricular mass indices are less strongly associated with impaired left ventricular function and, therefore, less useful in the identification of patients with chronic kidney disease who are at increased risk of heart failure.

## 2. Patients and Methods

### 2.1. Patients

This study was performed in accordance with the Helsinki Declaration, as revised in 2013. The University of Witwatersrand’s Human (Medical) Research Committee approved the protocol (protocol number: M15-08-43). Each patient provided informed written consent prior to participation. The study design was reported previously [27]. One hundred and three patients participated in this study. These patients included 62 non-dialysis and 41 dialysis patients. Exclusion criteria comprised active infection or/and cancer, previously diagnosed heart failure and having a Chronic Kidney Disease Epidemiology Collaboration-estimated glomerular filtration rate (eGFR) of ≥60 mL/min/1.73 m^2^.

### 2.2. Methods

We previously reported the applied methods were previously reported in another study [27,28,29]. Recorded baseline characteristics included demographic features, anthropometric measures, major traditional cardiovascular risk factors, non-traditional or renal cardiovascular risk factors, the use of cardiovascular drugs and erythropoietin stimulating agents and established cardiovascular disease. The latter factor comprised ischemic heart disease (acute myocardial infarction, percutaneous transluminal coronary angioplasty and coronary artery bypass graft), cerebrovascular disease (stroke and transient ischemic attack) and peripheral vascular disease (have), the presence of which was confirmed by a cardiologist, neurologist and vascular surgeon, respectively. All investigations were performed on a single day. In dialysis patients, this investigation was carried out on a day prior to a haemodialysis session. All patients were in sinus rhythm at the time of investigation. Mean arterial blood pressure for the peripheral waveform was determined electronically via the SphygmoCor device (see below) and using the below formula.


$$\begin{array}{cc} & T_{F}\\ & \sum_{P_{i}}\\ & i=T_{0}\\ MP= & \_\_\_\_\_\\ & \;n \end{array}$$


where T_0_ = the start of the waveform; T_F_ =the end of the waveform; P_i_ = pressure points; and, *n* = the number of pressure points.

Central systolic blood pressure was measured through utilizing a high-fidelity SPC-301 micromanometer (Miller instrument, Inc., Houston, TX, USA), which interfaced with a computer using SpygmoCor software, version 9.0 (AtCor Medical Pty. Ltd., West Ryde, NSW, Australia), as previously reported [26,27] After resting for 15 min in the supine position, arterial waveforms at the radial (dominant arm), carotid and femoral arteries were recorded for a time period of ten consecutive waveforms (heart beats). Calibration of the pulse wave was performed via manual measurement (auscultation) of the brachial blood pressure taken immediately prior to recordings. A validated generalized transfer function incorporated in the SphygmoCor software was used to convert the peripheral pressure waveform into a central aortic waveform. The results were discarded when systolic and diastolic variability in consecutive waveforms exceeded 5% or the amplitude of the pulse wave signal was less than 80 mV. All measurements were made by a single experienced observer (CR), who was unaware of the cardiovascular risk factor profiles of the patients. Brachial blood pressure was recorded in all patients. Technically sound measurements of the central pressure wave were obtained in 99 patients.

Echocardiography was performed in accordance with the American Society of Echocardiography convention [30] by employing a Philips CX50 POC Compact CompactXtreme Ultrasound System (Philips Medical Systems (Pty) Ltd., Andover, Massachusetts, USA) equipped with a 1.8–4.2 MHz probe that allowed for M-mode, 2-D, pulsed and tissue Doppler measurements, as previously described [28]. Patients were examined in the partial left decubitus position. We assessed left ventricular geometry and systolic (lateral s’ and midwall fractional shortening were measures of longitudinal and circumferential myocardial contractility, respectively, and ejection fraction was an index of chamber or pump function) and diastolic (e’ was a measure of active relaxation, and E/e’ was an index of passive relaxation and left ventricular filling pressure) functions.

Left ventricular dimensions were determined by measuring the left ventricular internal end diastolic and end systolic diameters and wall thicknesses (left ventricular septal and posterior wall thickness) in the parasternal long axis view via two-dimensional directed M-mode echocardiography. The Teichholz method was used to assess left ventricular end diastolic and systolic volumes, which were indexed to body surface area (LVEDVI-BSA and LVESVI-BSA). Stroke volume was determined from the difference between left ventricular end diastolic and systolic volumes, as evaluated upon employing the Z-derived method. Cardiac output was determined as stroke volume x heart rate. Left ventricular midwall fractional shortening was assessed using the previously reported formula, i.e., [(LVIDed + 0.5 Hed)—LVIDes + 0.5 Hes)]/LVIDed + 0.5 Hed), where LVID is left ventricular internal diameter, H is wall thickness (mean of septal + posterior wall thickness), ed is end diastole and es is end systole [29,31]. Left ventricular ejection fraction was calculated as [(left ventricular end diastolic volume—left ventricular end systolic volume)/left ventricular end diastolic volume] × 100.

Left ventricular mass (LVM) was determined using a standard formula and indexed to body surface area (LVMI-BSA) and height^1.7^ (LVMI-ht^1.7^). Left ventricular hypertrophy was identified as LVM-ht^1.7^ greater than 80 g/m^1.7^ for men and greater than 60 g/m^1.7^ for women. Stroke work (SW) was calculated as stroke volume × central systolic blood pressure x 0.014 and expressed in gram–meters/beat. Inappropriate left ventricular mass (iLVM) was determined from predicted LVM, where predicted LVM was calculated as 55.37 + (6.64 × height^2.7^) + (0.64 × [central systolic blood pressure × stroke volume × 0.014]) − (18.07 × sex), where male sex = 1 and female sex = 2. Inappropriate LVM was expressed as a percentage of actual LVM/predicted LVM. Increased inappropriate left ventricular mass was defined as inappropriate LVM if it recorded values above 128%. Traditionally determined left ventricular mass and inappropriate left ventricular mass could be calculated in 97 and 93 patients, respectively. Predicted LVM was then indexed to body surface area (pLVMI-BSA) and height^1.7^ (pLVMI-ht^1.7^). PLVMI-BSA and pLVMI-ht^1.7^ represented hemodynamic left ventricular remodeling [21,22,23]. Non-hemodynamic left ventricular remodeling was represented by inappropriate excess left ventricular mass indexed to body surface area (ieLVMI-BSA) and height^1.7^ (ieLVMI-ht^1.7^), which were defined as follows: measured LVMI-BSA—pLVMI-BSA and measured LVMI-ht^1.7^—pLVMI-ht^1.7^ [21,22,23].

Left ventricular relative wall thickness (RWT) was calculated as left ventricular diastolic posterior wall thickness × 2)/left ventricular end diastolic diameter. Increased relative wall thickness was determined as ≥0.45. Based on the LVMI and RWT measurements, geometric patterns were described as concentric remodeling (normal LVMI and increased RWT), concentric hypertrophy (increased LVMI and increased RWT) and eccentric hypertrophy (increased LVMI and normal RWT).

Transmitral flow patterns were recorded at the mitral valve leaflet tips using pulsed Doppler in the apical four-chamber view. The early (E) diastolic wave was measured from the mitral inflow velocity curve. Using tissue Doppler imaging, the velocities of longitudinal myocardial tissue shortening (s’) and early diastolic mitral annulus motion (e’) were measured by placing the cursor at the septal and lateral corners of the mitral annulus. The left ventricular filling pressure index (E/e’ ratio) was calculated as mitral E/the average of septal and lateral e’. An E/e’ > 14 was considered elevated.

Echocardiographic measurements were made by the same observer who performed the arterial function evaluation. Intra-observer echocardiographic measurement variability was low in our setting with Pearson’s correlation coefficients and variances (mean % difference (SD)) for left ventricular end-diastolic diameter, septal wall thickness, posterior wall thickness, E and e’ of 0.92, 0.72, 0.76, 0.88 and 0.93 (*p* < 0.0001 for all) and −0.41 (4.16), 0.45 (7.74), 1.74 (6.08), 0.16 (9.95) and −1.46 (8.58), respectively.

### 2.3. Data Analysis

Results were expressed as mean (SD), median (interquartile range) or proportions as appropriate. Non-normally distributed variables were logarithmically transformed prior to being entered into linear multivariable regression models.

Data among subgroups, including non-dialysis versus dialysis patients, patients with and without traditionally determined left ventricular hypertrophy and patients with appropriate and inappropriate left ventricular mass, were compared in age-, sex- and race-adjusted models.

Associations between left ventricular mass parameters and measures of diastolic and systolic function were assessed in confounder-adjusted models. Significant associations were re-evaluated via a sensitivity analysis among patients without established cardiovascular disease.

To assess the contribution of hemodynamic and non-hemodynamic components of cardiac remodeling to traditionally determined left ventricular mass indices associated with left ventricular function, we performed confounder-adjusted product of coefficient mediation analysis. Mediation analysis accounts for hierarchical causal structures [32,33,34,35,36].

The performance of left ventricular mass parameters in identifying patients with reduced ejection fraction was assessed using receiver operator characteristic (ROC) curve analysis. The optimal left ventricular mass parameter cut-off values required in predicting the presence of reduced ejection fraction were determined by calculating the Youden index.

The data were analyzed using IBM SPSS statistical program (version 27.0, IBM, Armonk, New York, USA) and Statistica 8.0 application package (version 14.0, TIBCO, Palo Alto, California USA).

## 3. Results

### 3.1. Recorded Baseline Characteristics in All, Dialysis and Non-Dialysis Patients

The baseline recorded characteristics in all, non-dialysis and dialysis patients are given in Table 1. Black patients were more often on dialysis, whereas white participants were more frequently not dialyzed. Systolic and mean blood pressure were larger in dialysis compared to non-dialysis patients. Haemoglobin concentrations were smaller, and erythropoietin stimulating agents and calcium channel blockers were more often used in dialysis than non-dialysis patients.

### 3.2. Central Pressures and Left Ventricular Structure and Function in All, Dialysis and Non-Dialysis Patients

Central pressures and left ventricular structure and function measures in all, non-dialysis and dialysis patients are shown in Table 2. Compared to non-dialysis patients, patients on dialysis had a larger central systolic blood pressure and LVMI-BSA, more frequent left ventricular hypertrophy indexed to height^1.7^ and eccentric hypertrophy, a larger ieLVMI-BSA and LEVDV-BSA and a smaller e’ and larger E/e’.

### 3.3. Baseline Recorded Characteristics in Patients with and without Traditionally Determined Left Ventricular Hypertrophy and Inappropriate Left Ventricular Mass

Baseline recorded characteristics in patients with and without traditionally determined left ventricular hypertrophy and inappropriate left ventricular mass are given in Table 3. Traditionally determined left ventricular hypertrophy was associated with a larger body mass index and systolic and mean blood pressure; the use of a larger number of antihypertensives; more frequent calcium channel blocker, beta blocker and erythropoietin-stimulating agent treatment; and more prevalent established cardiovascular disease. Compared to patients with appropriate left ventricular mass, patients with inappropriate left ventricular mass had a larger body mass index and weight and used beta blockers more frequently.

### 3.4. Central Pressures and Left Ventricular Structure and Function in Patients with and without Traditionally Determined Left Ventricular Hypertrophy and Inappropriate Left Ventricular Mass

Central pressures and left ventricular structure and function in patients with and without traditionally determined left ventricular hypertrophy and inappropriate left ventricular mass are shown in Table 4. Traditionally determined left ventricular hypertrophy was associated with increased inappropriate left ventricular mass, larger predicted and inappropriate excess left ventricular mass indices and LVEDVI-BSA, stroke volume, cardiac output and stroke work but reduced ejection fraction. Central systolic and pulse pressure were larger in patients with compared to those without traditionally determined left ventricular hypertrophy, but these differences did not reach significance (*p* = 0.05 and *p* = 0.06, respectively).

Compared to patients with appropriate left ventricular mass, those with inappropriate left ventricular mass had a larger LVMI-BSA and LVMI-ht^1.7^, a smaller predicted LVMI-BSA and larger inappropriate excess left ventricular mass indices, increased concentric remodelling, a larger LVEDV-BSA and reduced ejection fraction and midwall fractional shortening. Stroke volume, cardiac output and stroke work were also lower in patients with compared to those without inappropriate left ventricular mass but these differences did not reach significance.

### 3.5. Associations of Left Ventricular Mass Parameters with Diastolic and Systolic Function

As given in Table 5, in univariate analysis, traditionally determined left ventricular mass indices, inappropriate left ventricular mass and predicted and inappropriate excess left ventricular mass indices were not associated with diastolic function measures and lateral s’. Inappropriate left ventricular mass was inversely related to midwall fractional shortening. Traditionally determined left ventricular mass indices, inappropriate left ventricular mass and inappropriate excess left ventricular mass indices were inversely associated with ejection fraction. Predicted left ventricular mass indices were directly related to midwall fractional shortening and ejection fraction, whereas though inappropriate excess left ventricular mass indices were inversely related to midwall fractional shortening, these relationships did not reach significance.

Table 6 shows the independent associations between left ventricular mass parameters and diastolic and systolic function. None of the left ventricular mass parameters were related to diastolic function measures and lateral s’. Traditionally determined left ventricular mass indices tended to be (*p* = 0.05 for LVMI-BSA) or were (*p* = 0.04 for LVMI-ht^1.7^) inversely associated with ejection fraction. However, upon additional adjustment for inappropriate left ventricular mass or inappropriate excess left ventricular mass indices, traditionally determined left ventricular mass indices became directly associated with ejection fraction. In contrast, upon additional adjustment for predicted left ventricular mass indices, the inverse relationships between traditionally determined left ventricular mass indices and ejection fraction were strengthened. Additionally, left ventricular mass indices were strongly and inversely associated with midwall fractional shortening in the respective models. These phenomena are further illustrated in Figure 1.

As applied to results obtained in univariate analysis (Table 5), inappropriate left ventricular mass was strongly and inversely related to both midwall fractional shortening and ejection fraction. Also, these relationships remained unaltered upon additional adjustment for traditionally determined left ventricular mass indices. This context remained the case when we also adjusted unindexed left ventricular mass. Inappropriate left ventricular mass was more strongly associated with midwall fractional shortening (*p* = 0.03 and *p* = 0.04, respectively) and ejection fraction (*p* = 0.01 and *p* = 0.02, respectively) than LVMI-BSA and LVMI-ht^1.7^.

Predicted left ventricular mass indices were directly associated with midwall fractional shortening and ejection fraction.

Inappropriate excess left ventricular mass indices were inversely related to ejection fraction and tended (*p* = 0.07 to *p* = 0.08) to be inversely associated with midwall fractional shortening.

Dialysis status did not impact any of the left ventricular mass parameter–function relations (interaction *p* = 0.06 to *p* = 0.9). Hence, a stratified analysis of the left ventricular mass parameter–function relations based on dialysis status was not performed.

As given in Table 7, when predicted left ventricular mass indices and inappropriate excess left ventricular mass indices were entered into the same regression model, the former indices were directly related to ejection fraction, whereas the latter indices were inversely associated with ejection fraction.

Table 8 shows that the associations between traditionally determined left ventricular mass indices, inappropriate left ventricular mass and predicted or inappropriate excess left ventricular mass indices and midwall fractional shortening and ejection fraction remained consistent in a sensitivity analysis among the 83 patients who had no established cardiovascular disease. Associations between left ventricular mass parameters and midwall fractional shortening or ejection fraction in 83 CKD patients without cardiovascular disease are also shown in the below table.

The impact of predicted and inappropriate excess left ventricular mass indices on confounder-adjusted associations between traditionally determined left ventricular mass indices and ejection fraction, as determined in mediation analysis, are given in Table 9. Inappropriate excess left ventricular mass indices contributed between 74.0 and 76.9% to traditional left ventricular mass index–ejection fraction relationships. Contrastingly, predicted left ventricular mass indices attenuated the respective relationships by between 11.2 and 17.5%.

### 3.6. Confounder Adjusted and Mutually Independent Relationships of Left Ventricular Predicted and Inappropriate Excess left Ventricular Mass Indices with Traditionally Determined Left Ventricular Mass Indices

As shown in Table 10, predicted left ventricular mass indices and inappropriate excess left ventricular mass indices contributed were independent of one another in terms of the variation in traditionally determined left ventricular mass indices. Together with covariates, predicted left ventricular mass indices and inappropriate excess left ventricular mass indices explained 91% of the variation in traditionally determined left ventricular mass indices. Inappropriate excess left ventricular mass indices numerically contributed more than predicted left ventricular mass indices to traditionally determined left ventricular mass indices.

### 3.7. Performance of Traditionally Determined Left Ventricular Mass Indices, Inappropriate Left Ventricular Mass and Inappropriate Excess Left Ventricular Mass Indices in Identifying Patients with Reduced Ejection Fraction

The performances of traditionally determined left ventricular mass indices, inappropriate left ventricular mass and inappropriate excess left ventricular mass indices in identifying patients with reduced ejection fraction (<50%), as determined via ROC curve analysis, are given in Figure 2. Each of these left ventricular mass parameters was associated with reduced ejection fraction. Among these parameters, inappropriate left ventricular mass (area under the curve (AUC) (95% confidence interval (CI)) = 0.821 (0.722–0.919)), inappropriate excess LVMI-BSA (AUC (95% CI) = 0.805 (0.699–0.911)) and inappropriate excess LVMI-ht^1.7^ (AUC (95% CI) = 0.800 (0.692–0.907)) effectively discriminated between patients with and without reduced ejection fraction, with cut-off values and corresponding sensitivities and specificities of 169, 69 and 88%, 28 g/m^2^ and 87 and 71%, and 20 g/m^1.7^ and 87 and 67%, respectively. While inappropriate excess left ventricular mass indices performed similarly to inappropriate left ventricular mass in identifying patients with reduced ejection fraction, the AUC for the inappropriate left ventricular mass–reduced ejection fraction relationship was larger than that for the LVMI-BSA–ejection fraction (*p* = 0.04 for comparison) and LVMI-ht1.7–ejection fraction (*p* = 0.04 for comparison) associations.

Table 11 shows the cut–off and sensitivity or specificity values when sensitivity and specificity were set at 80% in ROC analysis for the left ventricular mass parameter–ejection fraction relationships.

## 4. Discussion

This study examined and compared for the first time the potential impact of traditionally determined left ventricular mass indices, inappropriate left ventricular mass and predicted and inappropriate excess left ventricular mass indices on diastolic and systolic function in patients with chronic kidney disease. Importantly in the present context, predicted left ventricular mass indices represent hemodynamic remodeling, whereas inappropriate left ventricular mass and inappropriate excess left ventricular mass indices are markers of non-hemodynamic remodeling. Left ventricular hypertrophy and inappropriate left ventricular mass were highly prevalent in the current chronic kidney disease cohort, which is consistent with findings reported in previous studies [1,2,3,4,5,23,24]. The main novel findings in this study are sixfold. Firstly, traditionally determined left ventricular hypertrophy and inappropriate left ventricular hypertrophy represent markedly different cardiac phenotypes in patients with chronic kidney disease. Secondly, none of the evaluated left ventricular mass parameters were associated with left ventricular diastolic function and longitudinal myocardial contractility in chronic kidney disease patients. Thirdly, inappropriate left ventricular mass and inappropriate excess left ventricular mass indices were more strongly associated with impaired systolic function than traditionally determined left ventricular mass indices. Fourthly, independent of one another, non-hemodynamic remodeling was strongly and inversely associated with left ventricular systolic function, whereas hemodynamic remodeling was equally strongly but directly associated with left ventricular systolic function. Fifthly, also independent of one another, both predicted and inappropriate excess left ventricular mass indices contributed to the variation in traditionally determined left ventricular mass indices. Inappropriate excess left ventricular mass indices numerically contributed more than predicted left ventricular mass indices to the variation in traditionally determined left ventricular mass indices. Sixthly, in ROC curve analysis, inappropriate left ventricular mass and inappropriate excess left ventricular mass indices, but not traditionally determined left ventricular mass indices, effectively discriminated between patients with and without decreased ejection fraction.

In the present study, the most prominent novel findings in patients with chronic kidney disease were uncovered via multivariate analysis, which assessed the potential relative impact of different left ventricular mass parameters on left ventricular circumferential myocardial contractility and chamber or pump function (Table 5,Table 6,Table 7,Table 8,Table 9 and Figure 1). Associations between left ventricular mass indices or inappropriate left ventricular mass and left ventricular diastolic and systolic function were previously reported in patients with hypertension [25] but, to the best of our knowledge, not in patients with chronic kidney disease. Herein, we found that non-hemodynamic left ventricular remodeling was strongly associated with impaired left ventricular systolic function, whereas hemodynamic left ventricular remodeling was also strongly but directly related to left ventricular systolic function. Although traditionally determined left ventricular mass indices tended to or were independently associated with reduced left ventricular systolic function, these relationships were far weaker and less consistent than those of non-hemodynamc left ventricular mass parameters with left ventricular systolic function. Most strikingly, upon additionally accounting for the effect of non-hemodynamic left ventricular remodeling parameters, traditionally determined left ventricular mass indices were directly related to left ventricular function. In contrast, upon additionally accounting for the effect of hemodynamic left ventricular remodeling, the inverse relationship between traditionally determined left ventricular mass indices and left ventricular systolic function was strengthened and, in fact, was as strong as the non-hemodynamic left ventricular remodeling–systolic function association. These results were reproduced via a sensitivity analysis, which comprised chronic kidney disease patients without cardiovascular disease. In a previous population study [15], inappropriate left ventricular mass was reported to be inversely associated with systolic function, being independent of absolute and indexed traditionally determined left ventricular mass. Our findings demonstrate that the same notion applies to patients with chronic kidney disease. However, whether traditionally determined left ventricular mass is associated with left ventricular systolic function independently of inappropriate and inappropriate excess left ventricular mass indices was not, to the best of our knowledge, previously reported.

In line with the above-mentioned findings, in separate models, hemodynamic and non-hemodynamic remodeling independently contributed to one another to a similar extent, albeit with opposing effects to left ventricular mass ejection fraction. Also, in mediation analysis, non-hemodynamic left ventricular remodeling accounted between 74.0 and 76.9% of the relationship between traditionally determined left ventricular mass indices and left ventricular systolic function, whereas for hemodynamic left ventricular remodeling, the respective association was between 11.2 and 17.5%.

Our findings suggest that in patients with chronic kidney disease, non-hemodynamic left ventricular remodeling parameters are more strongly associated with impaired left ventricular systolic function than traditionally determined left ventricular mass indices, because the latter approach represents both compensatory and adaptive hemodynamic left ventricular remodeling that aims to preserve wall stress in the face of increased pressure (concentric hypertrophy) and maintain a large stroke volume in the presence of increased volume load (eccentric hypertrophy), as well as cause adverse and maladaptive non-hemodynamic left ventricular remodeling [7]. Indeed, hemodynamic and non-hemodynamic left ventricular remodeling contributed, independently of one another, to the variation in traditionally determined left ventricular mass indices. Left ventricular non-hemodynamic remodeling contributed more than hemodynamic remodeling to the variation in traditionally determined left ventricular mass indices. This fact would explain why traditionally determined left ventricular mass indices are associated with impaired left ventricular systolic function, despite representing, in part, compensatory adaptive left ventricular remodeling, albeit to a much lesser extent than non-hemodynamic left ventricular mass parameters. Congruent with the above-mentioned results, in ROC analysis, non-hemodynamic left ventricular remodeling measures were more effective than traditionally determined left ventricular mass indices at identifying patients with reduced left ventricular function.

Systolic heart failure in hypertensive patients was previously found to be associated with reduced mysial collagen deposits and increased perivascular- and scar-related collagen deposits. Reduced mysial collagen can adversely impact systolic function through decreased support, geometric alignment and coordination of adjacent cardiomyocyte fascicle contraction; impaired synchrony and synergy of sarcomeres during contraction; and sliding displacement of cardiomyocytes [37]. Whether these abnormalities relate to inappropriate left ventricular mass merits further study.

This study revealed that e’ and E/e’ were similar in CKD patients with and without traditionally determined left ventricular hypertrophy and inappropriate left ventricular mass in patients with chronic kidney disease. Also, in both univariate and confounder-adjusted multivariable regression models, traditionally determined left ventricular mass indices, as well as inappropriate left ventricular mass and inappropriate excess left ventricular mass indices, were not associated with diastolic function. Left ventricular hypertrophy is mostly thought to contribute to diastolic dysfunction in patients with essential hypertension [38] and CKD [39,40]. However, it was recently shown that the development of diastolic dysfunction precedes that of left ventricular hypertrophy in patients [41,42,43] and rats [44] with essential hypertension. Similarly, in a mouse model of CKD [35], diastolic dysfunction preceded the development of left ventricular hypertrophy. Interestingly, this event was followed by the subsequent occurrence of fibrosis and increased expression of natriuretic peptides, which is suggestive of progressive heart failure. Importantly, in the present context, isolated diastolic function can cause pulmonary edema [43,45,46]. Recently reported data, together with our results, suggests that diastolic function evaluation via tissue Doppler imaging may be preferable to left ventricular mass parameters in assessing the risk of heart failure with preserved ejection fraction in CKD patients.

In the current investigation, predicted left ventricular mass indices, stroke volume, cardiac output and stroke work were each increased in patients with compared to without traditionally determined left ventricular hypertrophy, but were not increased in patients with compared to without inappropriate left ventricular hypertrophy. Also, in line with previous reports in patients with hypertension [25], inappropriate but not traditionally determined left ventricular hypertrophy was associated with concentric remodeling, as represented by left ventricular relative wall thickness. These findings document that traditionally determined left ventricular hypertrophy and inappropriate left ventricular hypertrophy represent different cardiac phenotypes. Indeed, among patients with traditionally determined and inappropriate left ventricular hypertrophy, the only association with baseline recorded statistics that was shared was that with body mass index.

The current study has limitations. Firstly, given its cross-sectional design, we cannot determine the direction of causality. Secondly, while we did not find differences in left ventricular mass parameter–left ventricular function relationships among non-dialysis and dialysis patients, inclusion of larger numbers of patients may reveal an impact of dialysis status on the respective associations. This possibility requires further investigation. Thirdly, systolic blood pressure was measured in the clinic. Blood pressure tends to fluctuate markedly over time in patients with chronic kidney disease [47]. The use of ambulatory blood pressure measurement upon calculating predicted left ventricular mass may, therefore, produce more robust results.

Our findings show the need for future larger and longitudinal studies to further elucidate the pathophysiological mechanisms that account for the impact of hemodynamic and non-hemodynamic remodeling on cardiac function in patients with chronic kidney disease.

## 5. Conclusions

Non-hemodynamic left ventricular remodeling is strongly associated with impaired left ventricular systolic function, whereas hemodynamic left ventricular remodeling relates strongly but directly to left ventricular systolic function. Independent of one another, hemodynamic and non-hemodynamic left ventricular remodeling associate with left ventricular systolic function (but in opposite directions) and directly associate with traditionally determined left ventricular mas indices. Non-hemodynamic remodeling contributes more than hemodynamic remodeling to traditionally determined left ventricular mass. Consequently, traditionally determined left ventricular mass indices are far less strongly associated with impaired left ventricular systolic function than inappropriate left ventricular mass and inappropriate excess left ventricular mass indices. Non-hemodynamic, but not hemodynamic, left ventricular mass parameters discriminates effectively between patients with and without reduced ejection fraction. Left ventricular mass parameters are not associated with impaired diastolic function in patients with chronic kidney disease. The most important practical implication in this study is that compared to traditionally determined left ventricular mass indices, inappropriate left ventricular mass and inappropriate excess left ventricular mass indices are substantially more useful and, therefore, preferable in identifying chronic kidney disease patients with systolic dysfunction, who are known to be at higher risk of risk of heart failure with reduced ejection fraction.

## Figures and Tables

**Figure 1 jcm-12-04211-f001:**
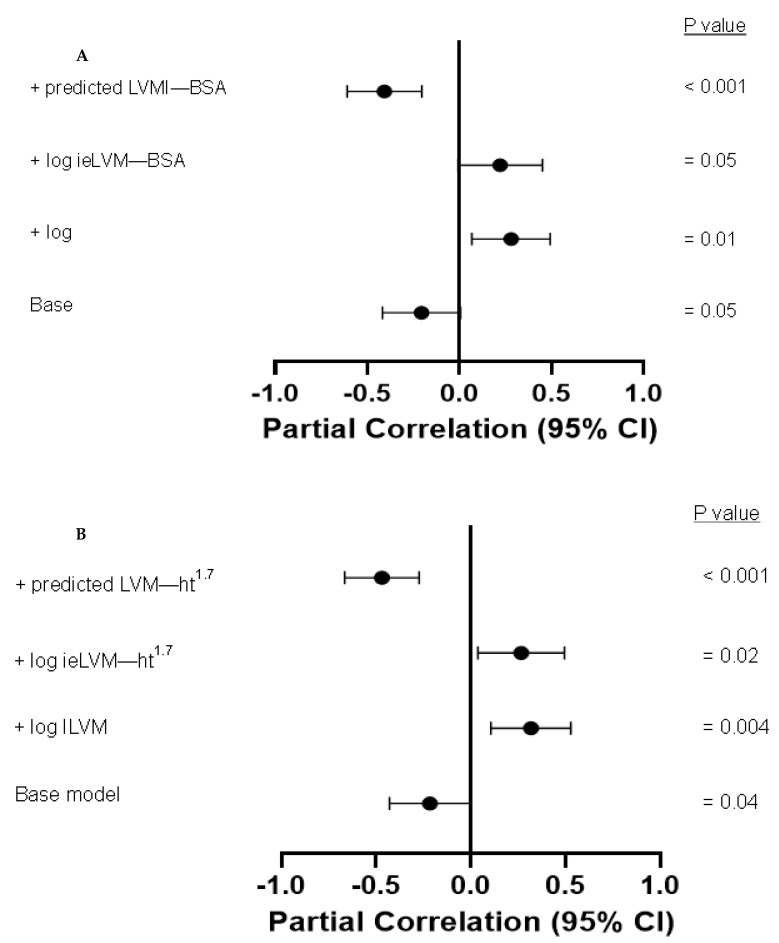
Associations between LVMI-BSA (**A**) or LVMI-ht^1.7^ (**B**) and ejection fraction both in base model and after adjusting for iLVM and inappropriate excess and predicted left ventricular mass indices. ILVM, inappropriate left ventricular mass; log, logarithmically transformed, i.e., inappropriate excess; LVMI-BSA, left ventricular mass indexed to body surface area; LVMI-ht^1.7^, left ventricular mass index indexed to height^1.7^.

**Figure 2 jcm-12-04211-f002:**
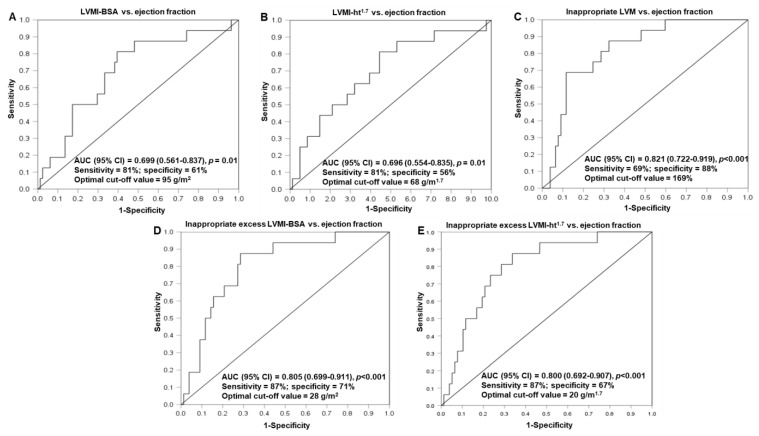
Receiver operator characteristic curves showing performance of LVMI–BSA (**A**), LVMI-ht^1.7^ (**B**), inappropriate LVM (**C**) and inappropriate excess LVMI-BSA (**D**) and LVMI-ht^1.7^ (**E**) in identifying patients with reduced ejection fraction (<50%). AUC, area under the curve; LVMI-BSA, left ventricular mass indexed to body surface area; LVM-ht^1.7,^ left ventricular mass indexed to height^1.7^; LVM, left ventricular mass.

**Table 1 jcm-12-04211-t001:** Baseline recorded characteristics in CKD patients.

Characteristics	All Patients (*n* = 103)	Non-Dialysis (*n* = 62)	Dialysis (*n* = 41)	*p*-Value
**Demographics**				
Age (years)	57.3 (14.5)	58.6 (14.1)	55.3 (15.1)	0.4
Female sex (%)	36.9	32.2	43.9	0.1
Black (%)	40.8	**29.0**	**58.5**	**0.004**
Asian (%)	29.1	33.9	21.9	0.1
White (%)	23.3	**33.9**	**7.3**	**0.006**
Mixed (%)	6.8	3.2	12.2	0.2
CKD duration	5.5 (4.1)	6.0 (4.5)	4.9 (3.3)	0.2
**Anthropometry**				
Body mass index (kg/m^2^)	27.6 (5.6)	27.8 (5.6)	27.2 (5.9)	0.5
Weight (kg)	78.7 (15.7)	80.7 (16.0)	75.7 (15.0)	0.2
Height (cm)	169 (10)	170 (9)	168 (11)	0.4
**Major traditional CV risk factors**				
Hypertension (%)	89.3	85.5	95.1	0.1
Systolic BP (mmHg)	141 (21)	**138 (20)**	**146 (21)**	**0.02**
Diastolic BP (mmHg)	83 (12)	81 (9)	85 (15)	0.1
Peripheral PP (mmHg)	58 (19)	57 (17)	61 (21)	0.2
Mean BP (mmHg)	102 (12)	**100 (11)**	**105 (14)**	**0.02**
Heart rate (beats/min)	75 (14)	74 (15)	76 (12)	0.4
Dyslipidemia (%)	79.6	86.0	69.4	0.06
Diabetes (%)	34	33.9	34.2	0.9
Smoking (%)	2.9	4.8	0.0	-
**Non-traditional CV risk factors**				
Dialysis duration (months)	-	-	36 (12–48)	-
Estimated GFR (mL/min/1.73 m^2^)	-	33 (17.9)	-	-
Phosphate (mmol/L)	1.2 (1.2–1.6)	1.2 (0.9–1.4)	1.4 (0.9–1.7)	0.1
Haemoglobin (g/dL)	11.6 (10.1–13.7)	**12.9(10.2–15.1)**	**10.8 (9.8–12.0)**	**<0.0001**
**Treatment**				
Antihypertensive agent use (%)	89.3	85.5	95.1	0.1
Antihypertensives (*n*)	2.1 (1.3)	2.0 (1.3)	2.3 (1.1)	0.2
ACEI/ARB use (%)	79.0	78.7	79.5	0.9
Calcium channel blocker use	43.6	33.9	59.0	**0.02**
Diuretic use	30.4	32.8	26.8	0.7
Beta blocker use	43.0	36.1	53.9	0.08
Alpha blocker use	22.2	21.3	23.6	0.7
Statin use (%)	62.0	46.7	51.3	0.1
ESA use (%)	46.6	**16.1**	**92.7**	**<0.001**
**Cardiovascular disease (%)**	19.4	20.9	17.1	0.8

Data were expressed as mean (SD), median (interquartile range) or proportions and analyzed in age-, sex- and race-adjusted regression models. Significant differences are shown in bold. CKD, chronic kidney disease; CV, cardiovascular; BP, blood pressure; GFR, glomerular filtration rate; ACEI, angiotensin converting enzyme inhibitors; ARB, angiotensin receptor blockers; ESA, erythropoietin-stimulating agents.

**Table 2 jcm-12-04211-t002:** Central systolic and pulse pressure, as well as left ventricular structure and function, in CKD patients.

Characteristics	All Patients (*n* = 103)	Non-Dialysis (*n* = 62)	Dialysis (*n* = 41)	*p*-Value
Central systolic BP (mmHg)	130 (19)	**127 (17)**	**135 (21)**	**0.01**
Central PP (mmHg)	44 (33–54)	42 (34–51)	49 (32–61)	0.1
LVM (g)	168 (133–230)	164 (130–236)	187 (144–229)	0.2
LVMI-BSA (g/m^2^)	93 (72–117)	**86 (71–110)**	**108 (80–120)**	**0.02**
LVMI-ht^1.7^ (g/m^1.7^)	68 (57–91)	64 (56–89)	77 (61–101)	0.05
LVH-ht^1.7^ (%)	44 (45.4)	**21 (35.6)**	**23 (60.5)**	**0.01**
ILVM (%)	134 (114–158)	134 (114–150)	142 (113–188)	0.3
ILVM increase (%)	50 (53.8)	32 (55.8)	18 (51.4)	0.8
PLVMI-BSA (g/m^2^)	71 (18)	68 (17)	74 (19)	0.05
PLVMI-ht^1.7^ (g/m^1.7^)	54 (13)	53 (13)	56 (14)	0.1
IeLVMI-BSA (g/m^2^)	20 (8–39)	20 (8–33)	**30 (9–51)**	**0.03**
IeLVMI-ht^1.7^ (g/m^1.7^)	15 (6–30)	15 (6–26)	20 (7–41)	0.05
LV relative wall thickness	0.36 (0.31–0.44)	0.37 (0.32–0.45)	0.33 (0.28–0.44)	0.1
Eccentric LVH (%)	36.1	**27.1**	**50.0**	**0.02**
Concentric LVH (%)	9.3	8.5	10.5	0.7
LVEDVI-BSA (ml/m^2^)	64 (46–86)	**61 (45–79)**	**77 (53–92)**	**0.03**
LV e’ (cm/s)	8.7 (2.7)	**9.0 (2.6)**	**8.1 (2.8)**	**0.01**
LV E/e’	10.0 (4.4)	**9.3 (4.2)**	**11.1 (4.6)**	**0.02**
LV E/e’ > 14	20	17.7	22.5	0.5
LV lateral wall s’ (cm/s)	8.7 (2.3)	8.9 (2.3)	8.2 (2.1)	0.1
LV midwall fractional shortening (%)	19.6 (4.7)	19.9 (4.7)	19.1 (4.9)	0.5
LV ejection fraction (%)	64 (14)	65 (14)	62 (14)	0.1
LV ejection fraction < 50% (%)	17.2	16.7	17.9	0.5
Stroke volume (mL/beat)	70 (24)	69 (24)	72 (24)	0.4
Cardiac output (L/min)	5.17 (1.92)	4.95 (1.87)	5.50 (1.98)	0.2
SW (gram-meters/beat)	140 (98–182)	135 (91–175)	154 (104–194)	0.2

Data were expressed as mean (SD), median (interquartile range) or proportions and analyzed in age, sex and race-adjusted regression models. Significant differences are shown in bold. CKD, chronic kidney disease; BP, blood pressure; PP, pulse pressure; LVM, left ventricular mass; LVMI-BSA, left ventricular mass indexed to body surface area; LVMI-ht^1.7^, left ventricular mass indexed to height^1.7^; LVH-ht^1.7^, left ventricular hypertrophy indexed to height^1.7^; ILVM, inappropriate left ventricular mass; PLVMI-BSA, predicted left ventricular mass indexed to body surface area; PLVMI-ht^1.7^ predicted left ventricular mass indexed to height^1.7^; IeLVMI-BSA, inappropriate left ventricular mass indexed to body surface area; IeLVMI-ht^1.7^, inappropriate excess left ventricular mass indexed to height^1.7^; LV, left ventricular; LVEDVI-BSA, left ventricular end diastolic volume indexed to body surface area; SW, stroke work.

**Table 3 jcm-12-04211-t003:** Baseline characteristics in CKD patients with and without traditionally determined left ventricular hypertrophy and inappropriate left ventricular mass.

Characteristics	No LVH (*n* = 53)	LVH (*n* = 44)	*p*-Value	Appropriate LVM (*n* = 43)	Inappropriate LVM (*n* = 50)	*p*-Value
**Demographics**						
Age (years)	56.8 (14.4)	57.0 (10.1)	1.0	56.0 (14.4)	57.4 (15.4)	0.6
Female sex (%)	37.7	36.4	0.9	37.2	34.0	0.8
Black (%)	39.6	43.2	0.7	39.5	42.0	0.8
Asian (%)	34.0	25.0	0.3	30.2	32.0	0.8
White (%)	20.7	22.7	0.8	23.3	18.0	0.4
Mixed (%)	5.7	9.1	0.5	7.0	8.0	0.9
CKD duration	6.0	5.2	0.4	5.8	5.6	0.7
**Anthropometry**						
Body mass index (kg/m^2^)	**26.2 (4.9)**	**28.3 (6.0)**	**0.04**	**25.8 (4.0)**	**28.3 (6.2)**	**0.03**
Weight (kg)	75.4 (13.8)	80.8 (17.1)	0.07	**74.2 (12.2)**	**81.0 (16.9)**	**0.03**
Height (cm)	170 (8)	169 (12)	0.6	170 (10)	169 (11)	0.9
**Major traditional CV risk factors**						
Hypertension (%)	92.5	88.6	0.5	90.7	90.0	0.9
Systolic BP (mmHg)	**136 (20)**	**146 (21)**	**0.03**	143 (22)	140 (20)	0.5
Diastolic BP (mmHg)	80 (10)	84 (13)	0.1	82 (11)	83 (12)	0.5
Peripheral PP (mmHg)	56 (20)	61 (18)	0.1	61 (21)	57 (17)	0.2
Mean BP (mmHg)	**99 (10)**	**105 (14)**	**0.03**	102 (12)	102 (13)	1.0
Heart rate (beats/min)	75 (16)	75 (13)	0.8	74 (15)	75 (15)	0.7
Dyslipidemia (%)	73.9	82.9	0.3	76.9	79.5	0.9
Diabetes (%)	34.0	34.1	1.0	27.9	38.0	0.4
Smoking (%)	3.8	2.3	0.8	2.3	4.0	0.6
**Non-traditional CV risk factors**						
Dialysis duration (months)	48 (24–48)	36 (21–39)	0.4	36 (18–48)	36 (18–36)	0.5
Estimated GFR (ml/min/1.73 m^2^)	34 (19)	35 (25)	0.3	37 (23)	33 (20)	0.9
Phosphate (mmol/l)	1.2 (1.0–1.4)	1.4 (0.9–1.6)	0.2	1.2 (0.9–1.6)	1.3 (0.9–1.5)	0.8
Haemoglobin (g/dl)	12.2 (10.5–13.7)	10.9 (9.6–13.4)	0.08	11.6 (10.2–13.8)	11.4 (9.7–13.7)	0.5
**Treatment**						
Antihypertensive agent use (%)	92.4	88.6	0.6	90.7	90.0	0.9
Antihypertensives (*n*)	**1.9 (1.0)**	**2.5 (1.4)**	**0.01**	2.0 (1.0)	2.3 (1.4)	0.3
ACEI/ARB use (%)	82.7	76.2	0.4	83.3	77.1	0.5
Calcium channel blocker use	34.0	61.9	**0.007**	41.9	47.9	0.6
Diuretic use	26.9	31.8	0.7	28.6	28.0	0.,9
Beta blocker use	18.9	59.5	0.002	26.2	56.3	0.005
Alpha blocker use	17.7	28.6	0.2	19.0	25.5	0.6
Statin use (%)	59.6	61.9	0.8	59.5	60.4	1.0
ESA (%)	**34.0**	**61.4**	**0.008**	46.5	44.0	0.8
**Cardiovascular disease (%)**	**11.3**	**25.0**	**0.03**	11.6	22.0	0.4

Data were expressed as mean (SD) or median (interquartile range), unless otherwise indicated, and analyzed in age-, sex- and race-adjusted regression models. Significant differences are shown in bold. CKD, chronic kidney disease; LVH, left ventricular hypertrophy; LVM, left ventricular mass; CV, cardiovascular; BP, blood pressure; GFR, glomerular filtration rate; ACEI, angiotensin converting enzyme inhibitors; ARB, angiotensin receptor blockers; ESA, erythropoietin stimulating agents.

**Table 4 jcm-12-04211-t004:** Central systolic and pulse pressure, as well as left ventricular structure and function, in CKD patients with and without traditionally determined left ventricular hypertrophy and inappropriate left ventricular mass.

Characteristics	No LVH (*n* = 53)	LVH (*n* = 44)	*p*-Value	Appropriate LVM (*n* = 43)	Inappropriate LVM (*n* = 50)	*p*-Value
Central systolic BP (mmHg)	126 (18)	134 (19)	0.05	131 (19)	128 (18)	0.3
Central PP (mmHg)	41 (33–51)	48 (34–61)	0.06	44 (35–54)	43 (33–53)	0.4
LVM (g)	**144 (155–167)**	**232 (191–284)**	**<0.001**	**148 (112–176)**	**208 (160–269)**	**<0.001**
LVMI-BSA (g/m^2^)	**76 (64–89)**	**118 (108–141)**	**<0.001**	**80 (64–93)**	**112 (84–130)**	**<0.001**
LVMI-ht^1.7^ (g/m^1.7^)	**59 (50–66)**	**94 (81–111)**	**<0.001**	**60 (46–72)**	**89 (66–109)**	**<0.001**
LVH-ht^1.7^ (%)	**0**	**100**	-	**18.6**	**68.0**	**<0.001**
ILVM (%)	**117 (104–135)**	**150 (134–201)**	**<0.001**	**113 (102–120)**	**154 (142–194)**	**<0.001**
ILVM increase (%)	**31.4**	**80.9**	**<0.001**	0	100	-
PLVMI-BSA (g/m^2^)	**66 (18)**	**77 (18)**	**0.003**	**75 (18)**	**67 (18)**	**0.03**
PLVMI-ht^1.7^ (g/m^1.7^)	**49 (12)**	**60 (13)**	**<0.001**	56 (12)	52 (14)	0.2
IeLVMI-BSA (g/m^2^)	**11 (3–19)**	**40 (25–63)**	**<0.001**	**8 (2–12)**	**38 (25–59)**	**<0.001**
IeLVMI-ht^1.7^ (g/m^1.7^)	**8 (2–15)**	**32 (20–56)**	**<0.001**	**6 (1–9)**	**29 (21–52)**	**<0.001**
LV relative wall thickness	0.39 (0.32–0.44)	0.34 (0.29–0.44)	0.5	**0.33 (0.29–0.41)**	**0.37 (0.33–0.46)**	**0.01**
Eccentric LVH (%)	**0**	**79.5**	-	**18.6**	**54.0**	**<0.001**
Concentric LVH (%)	**0**	**20.4**	-	**0**	**14.0**	-
LVEDVI-BSA (ml/m^2^)	**53 (44–67)**	**85 (71–108)**	**<0.001**	**64 (45–80)**	**72 (47–100)**	**<0.001**
LV e’ (cm/s)	9.0 (2.8)	8.6 (2.7)	0.4	9.1 (2.4)	8.6 (2.9)	0.5
LV E/e’	9.2 (4.1)	10.7 (4.5)	0.08	9.5 (4.0)	10.3 (4.8)	0.4
LV E/e’ > 14 (%)	15.1	25.0	0.2	11.6	28.0	0.07
LV lateral wall s’ (cm/s)	8.8 (2.3)	8.5 (2.3)	0.5	8.4 (7.6–10.1)	8.5 (7.1–10.1)	0.7
LV midwall fractional shortening (%)	20.1 (4.6)	18.9 (5.0)	0.3	**21.1 (4.3)**	**18.4 (4.9)**	**0.02**
LV ejection fraction (%)	**67 (13)**	**61 (14)**	**0.03**	**70 (10)**	**60 (15)**	**<0.001**
LV ejection fraction < 50% (%)	11.3	22.7	0.08	**4.7**	**28.0**	**0.01**
Stroke volume (mL/beat)	**62 (21)**	**80 (24)**	**<0.001**	74 (22)	68 (25)	0.2
Cardiac output (L/min)	**4.60 (1.64)**	**5.95 (1.98)**	**0.001**	5.45 (1.84)	4.97 (1.97)	0.2
SW (gram-meters/beat)	**111 (88–155)**	**165 (128–206)**	**<0.001**	151 (104–184)	129 (91–180)	0.2

Data were expressed as mean (SD), median (interquartile range) or proportions and analyzed in age-, sex- and race-adjusted regression models. Significant differences are shown in bold. CKD, chronic kidney disease; BP, blood pressure; PP, pulse pressure; LVM, left ventricular mass; LVMI-BSA, left ventricular mass indexed to body surface area; LVMI-ht^1.7^, left ventricular mass indexed to height^1.7^; LVH-ht^1.7^, left ventricular hypertrophy indexed to height^1.7^; ILVM, inappropriate left ventricular mass; PLVMI-BSA, predicted left ventricular mass indexed to body surface area; PLVMI-ht^1.7^ predicted left ventricular mass indexed to height^1.7^; IeLVMI-BSA, inappropriate left ventricular mass indexed to body surface area; IeLVMI-ht^1.7^, inappropriate excess left ventricular mass indexed to height^1.7^; LV, left ventricular; LVEDVI-BSA, left ventricular end diastolic volume indexed to body surface area; SW, stroke work.

**Table 5 jcm-12-04211-t005:** Univariate associations of left ventricular mass parameters with diastolic and systolic function in CKD patients.

LVM Parameter	E’		E/e’		Lateral s’		Midwall Fractional Shortening		Ejection Fraction	
	Partial R	*p*-Value	Partial R	*p*-Value	Partial R	*p*-Value	Partial R	*p*-Value	Partial R	*p*-Value
Log LVMI-BSA	−0.054	0.6	0.172	0.09	−0.093	0.4	−0.121	0.3	**−0.296**	**0.003**
Log LVMI-ht^1.7^	−0.091	0.4	0.186	0.07	−0.116	0.3	−0.172	0.1	**−0.294**	**0.003**
Log ILVM	−0.022	0.8	0.066	0.5	−0.049	0.6	**−0.413**	**<0.001**	**−0.461**	**<0.001**
PLVMI-BSA	−0.117	0.3	0.156	0.1	−0.108	0.3	0.235	0.05	0.121	0.2
PLVMI-ht^1.7^	−0.188	0.07	0.196	0.06	−0.148	0.2	0.158	0.2	0.114	0.3
Log ieLVMI-BSA	−0.142	0.2	0.196	0.07	−0.127	0.2	−0.217	0.09	**−0.409**	**<0.001**
Log ieLVMI-ht^1.7^	−0.159	0.1	0.203	0.06	−0.137	0.2	−0.230	0.07	**−0.402**	**<0.001**

Significant associations are shown in bold. CKD, chronic kidney disease; LVM, left ventricular mass; log, logarithmically transformed; LVMI-BSA, left ventricular mass indexed to body surface area; LVMI-ht^1.7^, left ventricular mass indexed to ht^1.7^; ILVM. Inappropriate left ventricular mass; PLVMI-BSA, predicted left ventricular mass indexed to body surface area; PLVMI-ht^1.7^, predicted left ventricular mass indexed to height^1.7^; ieLVMI-BSA, inappropriate excess left ventricular mass indexed to body surface area; ieLVMI-ht^1.7^, inappropriate excess left ventricular mass indexed to height^1.7^.

**Table 6 jcm-12-04211-t006:** Independent associations of left ventricular mass parameters with diastolic and systolic function in CKD patients.

LVM Parameter	E’			E/e’			Lateral s’			Midwall Fractional Shortening			Ejection Fraction		
	Partial R	*p*-Value	Model R^2^	Partial R	*p*-Value	Model R^2^	Partial R	*p*-Value	Model R^2^	Partial R	*p*-Value	Model R^2^	Partial R	*p*-Value	Model R^2^
Log LVMI-BSA	0.059	0.6	0.272	0.078	0.5	0.252	−0.051	0.6	0.147	−0.165	0.2	0.188	−0.206	0.05	0.205
Log LVMI-BSA ^a^	0.094	0.4	0.319	0.086	0.4	0.273	−0.040	0.7	0.168	0.253	0.05	0.374	**0.280**	**0.01**	**0.421**
Log LVMI-BSA ^b^	0.122	0.3	0.319	0.103	0.4	0.282	−0.035	0.8	0.166	0.097	0.5	0.209	**0.221**	**0.05**	**0.368**
Log LVMI-BSA ^c^	0.008	0.9	0.315	0.084	0.4	0.273	−0.063	0.6	0.168	**−0.349**	**0.006**	**0.362**	**−0.407**	**<0.001**	**0.413**
Log LVMI-ht^1.7^	0.019	0.9	0.270	0.102	0.4	0.255	−0.089	0.4	0.162	−0.188	0.1	0.196	**−0.216**	**0.04**	**0.211**
Log LVMI-ht^1.7 a^	0.012	0.9	0.316	0.125	0.3	0.279	−0.075	0.5	0.179	0.246	0.06	0.373	**0.317**	**0.004**	**0.455**
Log LVMI-ht^1.7 d^	0.046	0.7	0.315	0.152	0.2	0.294	−0.075	0.5	0.176	0.080	0.6	0.209	**0.266**	**0.02**	**0.406**
Log LVM-ht^1.7 e^	−0.006	1.0	0.316	0.083	0.5	0.279	−0.091	0.4	0.179	**−0.393**	**0.002**	**0.362**	**−0.470**	**<0.001**	**0.451**
Log ILVM	−0.021	0.8	0.316	0.055	0.6	0.267	−0.074	0.5	0.174	**−0.450**	**<0.001**	**0.332**	**−0.512**	**<0.001**	**0.394**
Log ILVM ^f^	−0.076	0.5	0.321	−0.033	0.8	0.273	−0.031	0.8	0.174	**−0.494**	**<0.001**	**0.386**	**−0.561**	**<0.001**	**0.461**
Log ILVM ^g^	−0.023	0.8	0.316	0.125	0.3	0.279	0.009	0.9	0.179	**−0.470**	**<0.001**	**0.373**	**−0.552**	**<0.001**	**0.455**
Log ILVM ^h^	−0.103	0.4	0.326	−0.009	0.9	0.270	−0.076	0.5	0.175	**−0.514**	**<0.001**	**0.396**	**−0.561**	**<0.001**	**0.460**
PLVMI-BSA	0.071	0.5	0.315	0.060	0.6	0.276	−0.026	0.8	0.164	**0.367**	**0.004**	**0.274**	**0.383**	**<0.001**	**0.296**
PLVMI-ht^1.7^	−0.007	0.9	0.316	0.108	0.3	0.274	−0.052	0.6	0.172	**0.315**	**0.01**	**0.246**	**0.378**	**<0.001**	**0.296**
Log ieLVMI-BSA	−0.037	0.7	0.309	0.126	0.3	0.274	−0.078	0.5	0.165	−0.238	0.08	0.201	**−0.373**	**<0.001**	**0.335**
Log ieLVMI-ht^1.7^	−0.048	0.7	0.313	0.132	0.3	0.278	−0.095	0.4	0.171	−0.245	0.07	0.204	**−0.389**	**<0.001**	**0.361**

CKD, chronic kidney disease; LVM, left ventricular mass; log, logarithmically transformed; LVMI-BSA, left ventricular mass indexed to body surface area; LVMI-ht^1.7^, left ventricular mass indexed to ht^1.7^; ILVM. Inappropriate left ventricular mass; PLVMI-BSA, predicted left ventricular mass indexed to body surface area; PLVMI-ht^1.7^, predicted left ventricular mass indexed to height^1.7^; ieLVMI-BSA, inappropriate excess left ventricular mass indexed to body surface area; ieLVMI-ht^1.7^, inappropriate excess left ventricular mass indexed to height^1.7^. ^a^ Additionally adjusted for log ILVM. ^b^ Additionally adjusted for log ieLVM-BSA. ^c^ Additionally adjusted for predicted LVMI-BSA. ^d^ Additionally adjusted for log ieLVM-ht^1.7^. ^e^ Additionally adjusted for predicted LVM-ht^1.7^. ^f^ Additionally adjusted for LVMI-BSA. ^g^ Additionally adjusted for LVMI-ht^1.7^. ^h^ Additionally adjusted for LVM.

**Table 7 jcm-12-04211-t007:** Mutually independent associations between predicted left ventricular mass indices or inappropriate excess left ventricular mass indices and ejection fraction in CKD patients.

Model	Partial R	*p*-Value	Model R^2^
Model 1			
PLVMI-BSA	**0.373**	**0.002**	
Log ieLVMI-BSA	**−0.331**	**<0.001**	**0.415**
Model 2			
PLVMI-ht^1.7^	**0.458**	**<0.001**	
Log ieLVMI-ht^1.7^	**−0.328**	**<0.001**	**0.454**

Data were analyzed via multiple linear regression models with adjustment for age, sex, race, weight (except for when predicted LVMI-BSA or ieLVMI-BSA were independent variables), heart rate, diabetes, mean blood pressure, use of erythropoietin stimulating agents and number of antihypertensive agents employed. Significant associations are shown in bold. CKD, chronic kidney disease; PLVMI-BSA, predicted left ventricular mass indexed to body surface areas; log, logarithmically transformed; PLVMI-height1.7, predicted left ventricular mass indexed to height1.7; ieLVMI_BSA, inappriate excess left ventricular mass indexed to body surface area; PLVMI-height^1.7^, predicted left ventricular mass indexed to height^1.7^; ieLVMI-ht^1.7^, inappropriate left ventricular mass indexed to height^1.7^.

**Table 8 jcm-12-04211-t008:** Associations between left ventricular mass parameters and midwall fractional shortening or ejection fraction in 83 CKD patients without cardiovascular disease.

LVM Parameter	Midwall Fractional Shortening			Ejection Fraction		
	Partial R	*p*-Value	Model R^2^	Partial R	*p*-Value	Model R^2^
Log LVMI-BSA	−0.235	0.09	0.258	**−0.285**	**0.01**	**0.199**
Log LVMI-BSA ^a^	0.238	0.09	0.439	**0.254**	**0.03**	**0.407**
Log LVMI-BSA ^b^	0.057	0.7	0.273	0.149	0.3	0.381
Log LVM-BSA ^c^	**−0.376**	**0.006**	**0.425**	**−0.447**	**<0.001**	**0.397**
Log LVMI-ht^1.7^	−0.240	0.08	0.260	**−0.276**	**0.02**	**0.195**
Log LVMI-ht ^1.7 a^	**0.280**	**0.04**	**0.459**	**0.307**	**0.01**	**0.430**
Log LVMI-ht1.7 ^d^	0.081	0.6	0.280	0.205	0.1	0.395
Log LVMI-ht1.7 ^e^	**−0.425**	**0.002**	**0.446**	**−0.489**	**<0.001**	**0.421**
Log ILVM	**−0.505**	**<0.001**	**0.413**	**−0.524**	**<0.001**	**0.371**
Log ILVM ^f^	**−0.531**	**<0.001**	**0.465**	**−0.529**	**<0.001**	**0.427**
Log ILVM ^g^	**−0.520**	**<0.001**	**0.459**	**−0.536**	**<0.001**	**0.430**
Log ILVM ^h^	**−0.452**	**<0.001**	**0.468**	**−0.522**	**<0.001**	**0.421**
PLVMI-BSA	**0.389**	**0.004**	**0.331**	**0.362**	**0.002**	**0.245**
PLVMI-ht^1.7^	**0.376**	**0.006**	**0.323**	**0.350**	**0.004**	**0.239**
Log ieLVMI-BSA	−0.284	0.05	0.270	**−0.453**	**<0.001**	**0.367**
Log ieLVMI-ht^1.7^	−0.291	0.05	0.275	**−0.452**	**<0.001**	**0.368**

Data were analyzed via multiple linear regression models with adjustment for age, sex, race, weight (except for when LVMI-BSA, pred LVMI-BSA or ieLVMI-BSA were independent variables), heart rate, diabetes, mean blood pressure, use of erythropoietin stimulating agents and number of antihypertensive agents employed. Significant associations are shown in bold. CKD, chronic kidney disease; LVM, left ventricular mass; log, logarithmically transformed; LVMI-BSA, left ventricular mass indexed to body surface area; LVMI-ht^1.7^, left ventricular mass indexed to ht^1.7^; ILVM. Inappropriate left ventricular mass;PLVMI-BSA, predicted left ventricular mass indexed to body surface area; PLVMI-ht^1.7^, predicted left ventricular mass indexed to height^1.7^; ieLVMI-BSA, inappropriate excess left ventricular mass indexed to body surface area; ieLVMI-ht^1.7^, inappropriate excess left ventricular mass indexed to height^1.7^. predicted; ie, inappropriate excess. ^a^ Additionally adjusted for log ILVM. ^b^ Additionally adjusted for log ieLVM-BSA. ^c^ Additionally adjusted for predicted LVMI-BSA. ^d^ Additionally adjusted for log ieLVM-ht^1.7^. ^e^ Additionally adjusted for predicted LVM-ht^1.7 f^ Additionally adjusted for LVMI-BSA. ^g^ Additionally adjusted for LVMI-ht^1.7^. ^h^ Additionally adjusted for LVM.

**Table 9 jcm-12-04211-t009:** Confounder-adjusted and mutually independent relationships between left ventricular predicted or inappropriate excess left ventricular mass indices and traditionally determined left ventricular mass indices in patients with CKD.

Model	Standardized β (5% to 95% CI)	*p*-Value	Model R^2^
Model 1 ^a^			
Predicted LVMI-BSA	0.561 (0.470 to 0.651)	<0.001	
Log ieLVMI-BSA	0.666 (0.594 to 0.738)	<0.001	0.914
Model 2 ^b^			
Predicted LVMI-ht^1.7^	0.544 (0.458 to 0.630)	<0.001	
Log ieLVMI-ht^1.7^	0.670 (0.599 to 0.741)	<0.001	0.917

Data were analyzed via multiple linear regression models with adjustment for age, sex, race, weight (except for when LVMI-BSA was the independent variables), heart rate, diabetes, mean blood pressure, use of erythropoietin stimulating agents and number of antihypertensive agents employed. Significant associations are shown in bold. CKD, chronic kidney disease; PLVMI-BSA, predicted left ventricular mass indexed to body surface areas; log, logarithmically transformed; PLVMI-height^1.7^, predicted left ventricularmass indexed to height1.7; ieLVMI_BSA, inappropriate excess left ventricular mass indexed to body surface area; PLVMI-height^1.7^, predicted left ventricular mass indexed to height^1.7^; ieLVMI-ht^1.7^, inappropriate left ventricular mass indexed to height1.7. ^a^ Traditionally determined LVMI-BSA was independent variable in this model. ^b^ Traditionally determined LVMI-ht^1.7^ was independent variable in this model.

**Table 10 jcm-12-04211-t010:** Confounder-adjusted and mutually independent relationships between left ventricular predicted or inappropriate excess left ventricular mass indices and traditionally determined left ventricular mass indices in patients with CKD.

Model	Standardized β (5% to 95% CI)	*p*-Value	Model R^2^
Model 1 ^a^			
Predicted LVMI-BSA	0.561 (0.470 to 0.651)	<0.001	
Log ieLVMI-BSA	0.666 (0.594 to 0.738)	<0.001	0.914
Model 2 ^b^			
Predicted LVMI-ht^1.7^	0.544 (0.458 to 0.630)	<0.001	
Log ieLVMI-ht^1.7^	0.670 (0.599 to 0.741)	<0.001	0.917

Data were analyzed via multiple linear regression models with adjustment for age, sex, race, weight (except for when LVMI-BSA was the independent variables), heart rate, diabetes, mean blood pressure, use of erythropoietin stimulating agents and number of antihypertensive agents employed. Significant associations are shown in bold. CKD, chronic kidney disease; PLVMI-BSA, predicted left ventricular mass indexed to body surface areas; log, logarithmically transformed; PLVMI-height^1.7^, predicted left ventricular mass indexed to height1.7; ieLVMI_BSA, inappropriate excess left ventricular mass indexed to body surface area; PLVMI-height^1.7^, predicted left ventricular mass indexed to height^1.7^; ieLVMI-ht^1.7^, inappropriate left ventricular mass indexed to height1.7. ^a^ Traditionally determined LVMI-BSA was independent variable in this model. ^b^ Traditionally determined LVMI-ht^1.7^ was independent variable in this model.

**Table 11 jcm-12-04211-t011:** Classification of patients with reduced ejection fraction with LVMI–BSA, LVMI–ht^1.7^, ILVM, inappropriate excess LVMI–BSA and inappropriate excess LVM–ht^1.7^ after setting sensitivity and specificity at 80%.

LVM Parameter	Cut–	Sensitivity (%)	Specificity (%)
LVMI-BSA (g/m^2^)	95	80	61
	117	69	80
LVMI-ht^1.7^ (g/m^1.7^)	68	80	56
	92	44	80
ILVM (%)	144	80	73
	150	69	80
IeLVMI-BSA (g/m^2^)	30	80	73
	38	63	80
IeLVMI-ht^1.7^ (g/m^1.7^)	22	80	71
	30	63	80

LVM, left ventricular mass; LVMI-BSA, left ventricular mass indexed to body surface area; LVMI-ht^1.7^, left ventricular mass indexed to height^1.7^; ILVM, inappropriate ventricular mass, Ie LVMI-BSA, inappropriate excess left ventricular mass indexed to body surface area; IeLVMI-height^1.7^, inappropriate left ventricular mass indexed to height^1.7^.

## Data Availability

All relevant data are contained within the manuscript.

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
