# Peer review of "Associations of Traditionally Determined Left Ventricular Mass Indices and Hemodynamic and Non-Hemodynamic Components of Cardiac Remodeling with Diastolic and Systolic Function in Patients with Chronic Kidney Disease"

_jcm, 2023, doi:10.3390/jcm12134211_

Round 1
Reviewer 1 Report
The scientific publication has been carried out in accordance with the standards of good practice. Illustrations and graphs have a high level of informativeness and significantly complement the work. One might get the impression that there are too many illustrations and tables, but they only support the conclusions drawn by the authors and make the text much easier to understand. The work will undoubtedly arouse interest in the professional expert environment and will also be useful as a reference in other scientific publications.
I recommend reviewing the tables and clarifying the explanations of the parameters and abbreviations specified in some of them.
Author Response
Reply to Reviewer 1
The scientific publication has been carried out in accordance with the standards of good practice. Illustrations and graphs have a high level of informativeness and significantly complement the work. One might get the impression that there are too many illustrations and tables, but they only support the conclusions drawn by the authors and make the text much easier to understand. The work will undoubtedly arouse interest in the professional expert environment and will also be useful as a reference in other scientific publications.
Reply:
Many thanks for your encouraging comments. We believe that your suggestions have helped us in enhancing the manuscript.
I recommend reviewing the tables and clarifying the explanations of the parameters and abbreviations specified in some of them.
Reply:
Indeed, there are many different parameters included in the present study. In accordance with your suggestions and to clarify the respective parameters, we have now spelled out the different abbreviations comprehensively in the footnotes of Tables 2,4,5,6,7,8,9,10 and11.

Reviewer 2 Report
The current manuscript titled: "Associations of Traditionally Determined Left Ventricular Mass Indices and Hemodynamic and Non-hemodynamic Components of Cardiac Remodeling with Diastolic and Systolic Function in Patients with Chronic Kidney Disease" represents an important analysis of evolving field of Cardiology and Nephrology.
In my opinion, these are the adjustments which should be made to increase the value of your manuscript:
1. In Abstract, after Background, please add study aim. Also, add conclusions.
2. It is not recommended to use figures in the Abstract section.
3. In Introduction chapter, please, add detailed information about the pathophysiological processes linking uremic cardiomyopathy and chronic kidney disease, and describe the relationship between chronic kidney disease and diastolic dysfunction.
4. In Methods chapter, please add peripheral vascular disease diagnosis criteria.
5. For this study, it is important to know the time period of onset of arterial hypertension and chronic kidney disease. It is also important to clarify the clinical status of the studied patients, including the degree of dyspnea, edema, etc.
6. It is not clear why there is such a high percentage of patients with ACEI/ARB use in the treatment of arterial hypertension (which is not the drug of choice in this case) and nothing is mentioned about calcium channel blockers, diuretics and centrally acting drugs.
7. In the Discussion section, there is not enough comparative information with other studies.
8. The conclusions largely repeat the Results section. In Conclusions section please highlight the practical implications of your study and its relevance to clinical practice.
9. Add future perspectives.
10. The manuscript contains some punctuation errors, please revise the text.
Moderate editing of English language required
Author Response
Reply to Reviewer 2:
The current manuscript titled: "Associations of Traditionally Determined Left Ventricular Mass Indices and Hemodynamic and Non-hemodynamic Components of Cardiac Remodeling with Diastolic and Systolic Function in Patients with Chronic Kidney Disease" represents an important analysis of evolving field of Cardiology and Nephrology.
Reply:
Many thanks for your encouraging comments. We believe that your suggestions have helped us in enhancing the manuscript.
In my opinion, these are the adjustments which should be made to increase the value of your manuscript:
- In Abstract, after Background, please add study aim. Also, add conclusions.
Reply:
We agree that this is needed. Accordingly, the first sentence of the Abstract was changed to: ‘We aimed to evaluate the extent to which different left ventricular mass parameters associate with left ventricular function in chronic kidney disease (CKD) patients.’ As the recommended word count is only
200 words, we have omitted the subheadings ‘Background’, ‘Patients and Methods’ and ‘Results and Conclusions’. The last sentence of the Abstract (‘Traditionally determined LVMIs are less strongly associated with impaired systolic function than non-hemodynamic remodeling parameters (p<0.04-0.01 for comparisons) because they represent both adaptive or compensatory and non-hemodynamic cardiac remodeling)’ comprises our main conclusion.
- It is not recommended to use figures in the Abstract section.
Reply: the respective Figure has been omitted.
- In Introduction chapter, please, add detailed information about the pathophysiological processes linking uremic cardiomyopathy and chronic kidney disease, and describe the relationship between chronic kidney disease and diastolic dysfunction.
Reply:
Thank you for these suggestions. Accordingly, we have now added the following sentences (with an additional reference (reference 4) in the first paragraph of the introduction of the revised manuscript on page 3): ‘The pathophysiological mechanisms that link chronic kidney disease with uremic cardiomyopathy include increased preload mediated by volume overload and anaemia, increased afterload due to arteriosclerosis and hypertension as well as a wide range of biochemical factors including chronic kidney disease bone and mineral disorder, renin-angiotensin-aldosterone and sympathetic nervous system overactivity, transforming growth-β, uremic toxins and endogenous cardiotonic steroids [1,2,3,4]. Diastolic dysfunction that can lead to heart failure with preserved ejection fraction is the most prevalent cardiac function abnormality in patients with chronic kidney disease [5].’
- In Methods chapter, please add peripheral vascular disease diagnosis criteria.
Reply:
We agree that this is needed. The diagnostic criteria for peripheral vascular disease included previous peripheral artery angioplasty and digital amputation due to ischemia. This has now been added to the first paragraph of Methods section in the second paragraph on page 5.
- For this study, it is important to know the time period of onset of arterial hypertension and chronic kidney disease. It is also important to clarify the clinical status of the studied patients, including the degree of dyspnea, edema, etc.
Reply:
We agree that these suggested data can enhance the manuscript. Accordingly, the duration of CKD is now given in Tables 1 and 3.
We did not record the duration of hypertension as the awareness of hypertension status among individuals in Africa is unreliably low (See Adeloye D and Basquill Catronia. Estimating the prevalence and awareness rates of hypertension in Africa: a systematic analysis. PLoS ONE 9(8): e104300.).
We could also not report on the clinical status of the patients as these were not recorded for the purpose of this study. The main reason for this was our understanding is that heart failure symptoms can reportedly be present in chronic kidney disease patients with and without heart failure and can be due to a range of factors including both volume overload and heart failure (see Adamska-Welnicka A et al. Chronic Kidney Disease and heart failure-everyday diagnostic challenges. Diagnostics 2021 Nov; 11(11):2164.). Nevertheless, we concur with the notion that the respective data could have been useful in the present context and should therefore be recorded in future studies.
- It is not clear why there is such a high percentage of patients with ACEI/ARB use in the treatment of arterial hypertension (which is not the drug of choice in this case) and nothing is mentioned about calcium channel blockers, diuretics and centrally acting drugs.
Reply:
We agree that ACEI/ARB were used frequently by the study participants. In this regard, hypertension is frequently severe and more refractory to treatment in Africa compared to high income countries. It is our understanding that the use and continuation of the respective agents has been recommended in patients with even advanced chronic kidney disease (See Ruggenenti P et al. ACE inhibitors to prevent end-stage renal disease: when to start and why possibly never to stop: a post hoc analysis of the REIN trial results: Ramipril Efficacy in Nephropathy. J Am Soc Nephrol 2001;12:2832-2837; Bhandari S et al. Renin-angiotensin system inhibition in advanced chronic kidney disease. N J Med 2022;387:2021-2032.).
In view of your comments, we now also report on the use of other antihypertensive agents including calcium blockers, diuretics, beta blockers and alpha blockers, this in Tables 1 and 3 and the respective preceding text on page 8 (third paragraph) and on page 10 (first paragraph and below Table 2). Centrally acting antihypertensive agents were not used in the study participants.
- In the Discussion section, there is not enough comparative information with other studies.
Reply:
We agree. While our main findings could not be compared to those in other studies as they were not previously reported in patients with chronic kidney disease, we have made the respective comparisons where feasible. Accordingly, we added the following sentences with appropriate references: ‘Left ventricular hypertrophy and inappropriate left ventricular mass were highly prevalent in the current chronic kidney disease cohort, which is consistent with reported findings in previous studies [1,2,3,4,5, 23,24].’ (in the first paragraph on page 27 and below Table11) and ‘In a previous population study [15], inappropriate left ventricular mass was reported to be inversely associated with systolic function independent of absolute and indexed traditionally determined left ventricular mass. Our findings demonstrate that the same applies in patients with chronic kidney disease. However, whether traditionally determined left ventricular mass is associated with left ventricular systolic function independent of inappropriate and inappropriate excess left ventricular mass indices has, to the best of our knowledge, not been reported previously.’ in the second paragraph on page 28.
- The conclusions largely repeat the Results section. In the Conclusions section please highlight the practical implications of your study and its relevance to clinical practice.
Reply:
We agree. We repeated major findings in the discussion to facilitate the interpretation of our results. We now highlighted the practical implications of the study and its relevance to clinical practice by including the following sentence in the fourth paragraph on page 30: ‘The most important practical implication of this study is that compared to traditionally determined left ventricular mass indices, inappropriate left ventricular mass and inappropriate excess left ventricular mass indices are substantially more useful and therefore preferable in identifying chronic kidney disease patients with systolic dysfunction who are known to be at higher risk of risk of heart failure with reduced ejection fraction.’.
- Add future perspectives.
Reply:
In accordance with this suggestion, we have now included the following sentence in the third paragraph on page 30: ‘Our findings call for future larger and longitudinal studies to further elucidate the pathophysiological mechanisms that account for the impact of hemodynamic and non-hemodynamic remodelling on cardiac function in patients with chronic kidney disease.
- The manuscript contains some punctuation errors, please revise the text.
Reply: Thank you. Punctuations were corrected throughout the manuscript.

Round 2
Reviewer 2 Report
I agree with the changes made by the authors, which greatly improved the quality of the manuscript.
Minor editing of English language required